# First-Trimester Influenza Infection Increases the Odds of Non-Chromosomal Birth Defects: A Systematic Review and Meta-Analysis

**DOI:** 10.3390/v14122708

**Published:** 2022-12-02

**Authors:** Ákos Mátrai, Brigitta Teutsch, Alex Váradi, Péter Hegyi, Boglárka Pethő, Akari Fujisawa, Szilárd Váncsa, Balázs Lintner, Zsolt Melczer, Nándor Ács

**Affiliations:** 1Department of Obstetrics and Gynecology, Semmelweis University, 1085 Budapest, Hungary; 2Centre for Translational Medicine, Semmelweis University, 1085 Budapest, Hungary; 3Faculty of Health Sciences, University of Pécs, 7621 Pécs, Hungary; 4Institute for Translational Medicine, Medical School, University of Pécs, 7621 Pécs, Hungary; 5Division of Pancreatic Diseases, Heart and Vascular Center, Semmelweis University, 1085 Budapest, Hungary; 6Faculty of Medicine, Semmelweis University, 1085 Budapest, Hungary

**Keywords:** viral infection, pregnancy, congenital malformations

## Abstract

Viral infections during pregnancy raise several clinical challenges, including birth defects in the offspring. Thus, this systematic review and meta-analysis aims to prove and highlight the risk of birth defects after first-trimester maternal influenza infection. Our systematic search was performed on 21 November 2022. Studies that reported maternal influenza infection in the first trimester and non-chromosomal congenital abnormalities were considered eligible. We used odds ratios (OR) with 95% confidence intervals (CIs) to measure the effect size. Pooled ORs were calculated with a random effects model. Heterogeneity was measured with I² and Cochran’s Q tests. We found that first-trimester maternal influenza was associated with increased odds of developing any type of birth defects (OR: 1.5, CI: 1.30–1.70). Moreover, newborns were more than twice as likely to be diagnosed with neural tube defects (OR: 2.48, CI: 1.95–3.14) or cleft lip and palate (OR: 2.48, CI: 1.87–3.28). We also found increased odds of developing congenital heart defects (OR: 1.63, CI: 1.27–2.09). In conclusion, influenza increases the odds of non-chromosomal birth defects in the first trimester. The aim of the present study was to estimate the risk of CAs in the offspring of mothers affected by first-trimester influenza infection.

## 1. Introduction

The recent outbreak of COVID-19 pneumonia, caused by SARS-CoV-2, has highlighted the role of viral infections during pregnancy as well [1,2]. The importance of these infections will probably increase as we face growing risks of pandemics, which may affect the pregnant mother and the fetus [3]. Viral infections during pregnancy raise several clinical challenges, including adverse pregnancy outcomes and birth defects in the offspring [4].

Influenza is an acute infectious disease caused by influenza A, B or C viruses. Most cases occur during epidemic outbreaks, generally between December and March. The latency period between infection and the manifestation of symptoms is about 48 h. Healthy individuals usually recover in 3–7 days; however, influenza may be followed by secondary bacterial infections of the respiratory system [5]. Pregnant women are among the groups that are at increased risk of complications [6].

Influenza may occur in pregnant women. Organogenesis takes place in the first trimester, so any environmental effect that occurs during this period may affect the development of the embryo. However, the role of influenza viruses is debated in the origin of congenital abnormalities (CAs). Some epidemiologic studies have shown a small increase in CAs in general or in some specific CAs (e.g., heart defects, esophageal atresia, anencephaly), while others did not find any increase of CAs after influenza epidemics [7,8,9,10,11,12]. However, data from these reports are difficult to assess because of lacking serologic proof of the microbial agents, unknown time of infection during pregnancy, restricted numbers of CAs and inadequate controls. One meta-analysis [13] on this topic also highlighted an association between maternal influenza and non-chromosomal birth defects. However, the included studies were not restricted to first-trimester infection It is not clear whether there is a direct link between influenza infection and the development of birth defects or it exerts an indirect teratogenic effect. High fever associated with influenza was assumed to play a causative role in this pathologic process [14].

Our study hypothesized that influenza in the first trimester increases the likelihood of developing congenital abnormalities. Thus, the present study aimed to estimate the risk of CAs in the offspring of mothers affected by influenza during the first trimester of the pregnancy. Worldwide, 3–5% of newborns are affected by a congenital anomaly [14]. These birth defects are to a great extent responsible for infant mortality [15]. The prevention of CAs is considered a public health priority due to their medical and financial consequences. Therefore, finding the role of influenza infection in the development of CAs is also of significant importance.

## 2. Materials and Methods

The systematic review and meta-analysis is reported according to the PRISMA (Preferred Reporting Items for Systematic Reviews and Meta-Analyses) 2020 Statement (see Appendix A) [16] and the MOOSE Checklist (see Appendix A) [17]. We followed the recommendations of the Cochrane Handbook for Systematic Reviews of Interventions [18]. The protocol was prospectively registered in PROSPERO (International Prospective Register of Systematic Reviews) under the registration number CRD42021283210 (https://www.crd.york.ac.uk/prospero, accessed on 19 October 2021). There was no protocol deviation.

### 2.1. Systematic Search

The systematic search was performed in three major medical databases: MEDLINE (via PubMed), Cochrane Library (CENTRAL) and Embase on 20 October 2021, and we reran our search on 21 November 2022. The search key contained the following main components: influenza, pregnancy, birth defect or congenital anomalies. The detailed queries are shown in the Appendix A. There were no filters or other restrictions used during the search.

### 2.2. Selection and Eligibility Criteria

We used EndNote 20 (Clarivate Analytics, Philadelphia, PA, USA) to select the retrieved articles. After removing duplications, two independent authors screened the library separately by title and abstract, then by full text (Á.M., A.F.). We calculated Cohen’s kappa coefficient (κ) after each selection process to measure interrater reliability [19]. Disagreements were resolved after each step by a third author (B.T.).

We used the population-exposure-outcome (PEO) framework to structure our research question. Case-control studies and cohort analyses, including pregnant women (P), who were investigated for influenza infection (E) during the first trimester of pregnancy, were found eligible. The outcomes of interest were all types of non-chromosomal birth defects (O).

The exclusion criteria concerned conference abstracts, articles reporting on common cold or fever instead of influenza, and influenza before or after the first trimester. To identify all relevant articles on the topic, the reference list of eligible full texts, and studies that cited the key articles were also checked. If the articles were not written in English, we sought the help of a translator. We contacted the authors if there was no available text or data from an article. When the study population was overlapping for an outcome, we included the study with a larger sample size.

### 2.3. Data Extraction

Two authors independently (Á.M., A.F.) extracted the data. We created a standardized data collection Excel sheet (Office 365, Microsoft, Redmond, WA, USA), which included first author, year of publication, title of the study, digital object identifier (DOI) number, study design, study type, study period, country, number of the participant centers, demographic characteristics of the included patients, number of participants, population characteristics of the study, data about the exposure (definition, time), raw data regarding the outcomes and unadjusted and adjusted results separately. A third author (B.T.) resolved the disagreements between the two authors.

### 2.4. Data Synthesis and Analysis

Qualitative data synthesis was done in the R programming language (R Core Team, 2012, Vienna, Austria, R version 4.2) using the meta v5.5.0 [20] and dmetar v0.0.9000 packages. A meta-analysis was performed if there were a minimum of three studies for one outcome. Where pooling effect sizes could not be carried out, we only visualized results on forest plots.

When possible, pooled odds ratios (ORs) with 95% confidence intervals (CIs) were calculated from 2 by 2 tables raw data or from crude ORs for the specific outcomes. Where possible, pooled adjusted ORs (AORs) were also estimated. A *p*-value less than 0.05 was considered statistically significant. Since we anticipated considerable between-study heterogeneity, a t random-effects model was applied in case of raw data using the Mantel-Haenszel [21,22] method by metabin function from meta package, and we applied the Paule-Mandel method to estimate the between-study variance [23]. Other cases we used already calculated crude or adjusted ORs and their standard errors for pooling with restricted maximum likelihood methods [24] with the help of metagen function. Heterogeneity was tested with I^2^ and Cochran Q tests; *p* < 0.1 indicated significant heterogeneity [25].

Following the recommendations of IntHout et al. (2016), where applicable, we also reported the prediction intervals (i.e., the expected range of effects of future studies) of the pooled estimates.

There were less than 10 studies for each outcome; therefore, a publication bias assessment could not be performed.

### 2.5. Risk of Bias Assessment and Quality of Evidence

The risk of bias assessment was performed by two review authors (Á.M., B.P.) independently using the Quality in Prognosis Studies (QUIPS) tool [26]. Disagreements were resolved by a third review author (B.T.). We visualized our results in figures with the Risk-of-bias VISualization (robvis) tool [27]. The overall assessment was considered low-risk if only low-risk bias domains existed. If one domain was at high risk or three out of six domains were at moderate risk, the overall result was reported as having a high risk of bias. The quality evaluation of the evidence was performed following the recommendations of the Grades of Recommendation, Assessment, Development, and Evaluation (GRADE) workgroup. We prepared the summary of findings tables with the GRADEPro Guideline Development Tool [28].

## 3. Results

### 3.1. Search and Selection

The systematic search yielded 10,450 records. After the duplication removal process, 6229 records remained. All records were checked by title and abstract, and 195 by full text. After overlapping populations were excluded, 14 articles [8,29,30,31,32,33,34,35,36,37,38,39,40] were included in the qualitative and quantitative synthesis. The selection process and Cohen’s kappas are shown in Figure 1.

### 3.2. Basic Characteristics of Included Studies

Baseline characteristics of the enrolled analyses are shown in Table 1. We included 13 case-control studies [8,29,30,31,32,34,35,36,37,38,39,40,41], and one retrospective cohort analysis [33]. The overall study period of the articles is placed between 1960 and 2020. The largest database included in the meta-analysis was the Hungarian Case-Control Surveillance of Congenital Abnormalities [29,34]. The diagnosis of influenza infection was in most cases based on the symptoms of the disease by using retrospective questionnaires. There was only one study in which the disease was confirmed by a polymerase chain reaction (PCR) test [33]. In most of the studies, the age of the mothers was between 20–34 years. The summary table of the outcomes of the included studies was shown also in the Appendix A.

### 3.3. Non-Chromosomal Congenital Malformations

Only one article reported a total number of non-chromosomal congenital malformations [30]. On the basis of their results, first-trimester maternal influenza was associated with 1.5 times increased odds (CI: 1.30–1.70) of developing all types of non-chromosomal birth defects.

### 3.4. Influenza Increases the Odds of Neural Tube Defects

Four eligible articles [34,35,36,38] reported on neural tube defects (NTDs). Our results highlighted that first-trimester maternal influenza infection increased the odds of developing NTDs (OR: 2.48, CI: 1.95–3.14; I^2^ = 0%, CI: 0–85%; Figure 2).

Pooled results from three articles [34,35,36] showed that influenza was associated with more than two times increased odds for the development of spina bifida (OR: 2.22, CI: 1.58–3.12; I^2^ = 0%, CI: 0–90%; Figure 3).

Two studies provided data for the assessment of encephalocele. Czeizel et al. [34] reported an almost twofold increase (OR: 1.70 CI: 0.69–4.16), whereas Lynberg et al. [36] reported an almost fourfold increase (OR: 3.94 CI: 1.34–11.58) in the development of this outcome. Granroth et al. [35] and Czeizel et al. [34] reported data about the development of microcephaly and hydrocephalus after first-trimester maternal influenza. They found a twofold increase in odds for microcephaly (Granroth et al. [35]: OR: 2.05 CI: 0.18–23.55; Czeizel et al. [34]: OR: 2.30 CI: 0.39–13.66) and a two-and-a-half-fold increase in odds for hydrocephalus (Granroth et al. [35]: OR: 2.61 CI: 0.99–6.86; Czeizel et al. [34]: OR: 2.30 CI: 1.12–4.70). When assessing spina bifida of the lower and upper part of the spine, Park et al. [38] reported an almost fourfold increase (OR: 3.73 CI: 1.72–8.09) and a four-and-a-half-fold increase (OR: 4.57 CI: 1.81–11.52) after first-trimester maternal influenza infection, respectively. These results are visualized in the Appendix A.

### 3.5. Oral Clefts

The association between first-trimester maternal influenza infection and cleft lip and palate was reported in four articles [34,39,40,41]. Overall results showed an almost 2.5-fold increase in the development of cleft lip and palate (OR: 2.48, CI: 1.87–3.28; I^2^ = 19%, CI: 0–88%; Figure 4). Also, Czeizel et al. found an association between the assessed risk factor and cleft lip (OR: 2.40, CI: 1.42–4.06). Moreover, L. Ács et al. found an even stronger association between the assessed exposure and cleft palate (OR: 2.95, CI: 1.75–4.95).

### 3.6. Congenital Heart Defects

Several studies [33,36,37,42] provided data for congenital heart defects. Li et al. [8] found 1.62 times higher odds (CI: 1.17–2.24), and Czeizel et al. [34] reported 1.90 times higher odds (CI: 1.50–2.40) for developing all types of congenital heart defects. Several studies also found a significant association between influenza and aortic, conotruncal, septal and ventricular septal defects. Odds ratios for the subtypes of congenital heart defects are presented in Table 2.

Three studies [8,32,37] reported adjusted odds ratios (AORs). The pooled results showed a more than 1.5 times increase in the odds of developing congenital heart defects (AOR: 1.63, CI: 1.27–2.09; I^2^ = 0%, CI: 0–90%; Figure 5).

We also found a positive association between influenza and specific types of congenital heart defects when the analysis was performed with multiple logistic regression. The results of the studies showed an almost fourfold increase in the development of aortic coarctation, an almost threefold increase in the development of left-sided obstructive heart defects and a two-and-a-half-fold increase in the development of right-sided obstructive heart defects. The findings of the studies showed an almost threefold increase in the development of atrial septal defects, a twofold increase in the development of ventricular septal defects, more than one-and-a-half-fold increase in the development of conotruncal defects. These results are shown in Table 3.

### 3.7. Other Types of Birth Defects

In our systematic review, several studies reported data on the association of influenza and other specific types of birth defects. Aro et al. [31] and Czeizel et al. [34] found a positive association between first-trimester maternal influenza infection and limb reduction defects (OR: 1.98, CI: 1.13–3.46; OR: 2.30, CI: 1.30–4.08, respectively). These findings are shown in the Appendix A.

We also found an association between maternal influenza and eye anomalies. Busby et al. [33] reported 1.26-fold odds (OR: 1.26, CI: 1.02–1.56), whereas Czeizel et al. [34] reported 1.30-fold odds (OR: 1.30, CI: 0.47–3.58) for the development of congenital malformations of the eyes. These findings are also shown in the Appendix A.

Several additional congenital malformations were found to be associated with first-trimester influenza infection. Details of the results can be found in Appendix A.

### 3.8. Risk of Bias Assessment and Level of Evidence

The evaluation was based on six criteria. Every domain was scored as carrying low, moderate or high risk of bias for all the included outcomes. The results of the risk of bias assessment are presented in the Appendix A. As the prognostic factor measurement was not carried out with PCR and the chances of recall bias were high, the overall risk of bias was high in most of the included studies. Also, the quality of evidence was very low for most of the results (Appendix A).

## 4. Discussion

The importance of viral infections will increase as we face growing risks of pandemics, which may affect the pregnant mother and the fetus [42]. Viral infections during pregnancy raise several clinical challenges, including adverse pregnancy outcomes and birth defects in the offspring [3]. During embryogenesis, the development of different organs and organ systems occurs one after the other and parallel to others. Almost every organ system develops in the first 12 weeks of pregnancy, so any effect on the embryo during this period may significantly influence the development of birth defects.

This systematic review and meta-analysis evaluated the effects of influenza infection in the first trimester of pregnancy on the development of non-chromosomal birth defects.

We know from a previous study by Ács et al. [30] that the odds of developing all types of non-chromosomal birth defects are 1.5 times higher if influenza is present in the first trimester (OR: 1.40; CI: 1.30−1.60). Oster et al. [43] got the same association (OR: 1.11; CI: 0.91−1.35) between influenza and birth defects in the USA. On the basis of the results of this meta-analysis, we could also confirm the association between first-trimester maternal influenza and the development of non-chromosomal congenital malformations. Moreover, our results showed increased odds of developing three major types of CAs, namely congenital heart defects, neural tube defects and oral clefts after influenza infection in the first three months of pregnancy.

Neural tube defects are congenital brain, spine, or spinal cord abnormalities. Many studies [44,45,46] have investigated the pathogenesis of NTDs. Among the epidemiology and risk factors chromosomal anomalies (trisomy 13, 18 and triploidy) represent less than 10% of all NTDs [47]. Maternal diabetes, obesity, maternal hyperthermia (use of sauna, hot water tube, and fever), medication (valproate) and low folate intake were also considered as risk factors for developing NTDs [48]. The quantitative and qualitative synthesis of our findings in the present study confirmed the association between influenza and neural tube defects [34,35,36]. Their significance is outstanding as they are amongst the most common causes of genetic abortions. Most importantly, we can decrease the development of these congenital malformations with folic acid supplementation [49].

Our findings also confirmed the association between the development of cleft lip and palate and first-trimester maternal influenza. Orofacial clefts are common birth defects, estimated at 1.5 per 1000 live births worldwide [50]. Newborns with these anomalies have feeding difficulties and often develop conductive hearing loss and speech problems [51]. Environmental agents and genetic factors were also linked with the development of these birth defects. Moreover, orofacial clefts were also associated with teratogenetic agents, such as alcohol or anticonvulsant drugs. However, folic acid deficiency can also increase the chances for these birth defects [52,53]. In a population-based case-control study, L. Ács et al. [29] confirmed that influenza is a risk factor for developing cleft palate like other lifestyle factors (gender, birthweight and smoking) or illnesses. Waller et al. [7] examined common cold/flu during the periconceptional period, and they assessed that the odds of developing a cleft lip was 1.23 (CI: 1.05−1.45).

Congenital heart defects are among the most prevalent birth defects, and they remain an important causative factor of neonatal and infant deaths. The risk of these birth defects was associated with low maternal education, pregestational diabetes, self-reported maternal clotting disorders, prescriptions for anticlotting medication [43] and maternal influenza infection [43,54]. Our results could prove the association between influenza infection and congenital heart disease [8,34].

Organogenesis takes place during the first 12−13 weeks of pregnancy. This time-frame is the so called critical period for the development of birth defects. Therefore, our meta-analysis focused only on the first three months of pregnancy. We identified several studies, which analyzed the periconceptional period (i.e., three to one months before pregnancy and the first trimester) [8,44,55]. These results do not differ significantly from the results we have obtained. Further studies have shown that if fever during an influenza infection was relieved effectively with antipyretic drugs, malformations are less likely to develop [30]. Several studies have also shown that taking folic acid and maternal vitamins during the first trimester of pregnancy can reduce the odds of developing birth defects [30,56].

### 4.1. Strengths and Limitation

To our knowledge, this is the first meta-analysis that investigated the association between maternal influenza and all types of non-chromosomal abnormalities during the first trimester. We followed our protocol, which was registered previously. Our investigation covered a long study period with a large number of cases. The articles were from all over the world, giving a comprehensive picture of the disease and its effects.

In consideration of the limitations of this analysis, most studies defined the diagnosis of influenza based on the symptoms (headache, fever, running nose and muscle pain) and reported data that were retrospective and self-reported. Thus, the risk of bias was high because of the prognostic factor measurement. Most of the studies included were case-control investigations, where a high percentage of recall bias is to be expected. We could include only one study in which a serological test identified the disease. Because of the rigorous methodology and timing, there were few cases for one outcome.

### 4.2. Implications for Practice and Research

Translating scientific results into everyday practice has crucial importance [56]; therefore, we should make efforts to prevent influenza during pregnancy and highlight the importance of vaccination against flu and antipyretic therapy. We suggest that if influenza is presented in the critical pregnancy period, great emphasis should be placed on prenatal screening. Additional prospective observational studies should be performed with the use PCR to confirm first-trimester influenza infection.

## 5. Conclusions

In conclusion, our systematic review found that influenza in the first trimester is associated with non-chromosomal birth defects, primarily congenital heart defects, neural tube defects and oral clefts. Therefore, influenza prevention by vaccination before or during pregnancy is highly recommended. Moreover, if influenza has already manifested itself, proper antipyretic treatment should be started.

## Figures and Tables

**Figure 1 viruses-14-02708-f001:**
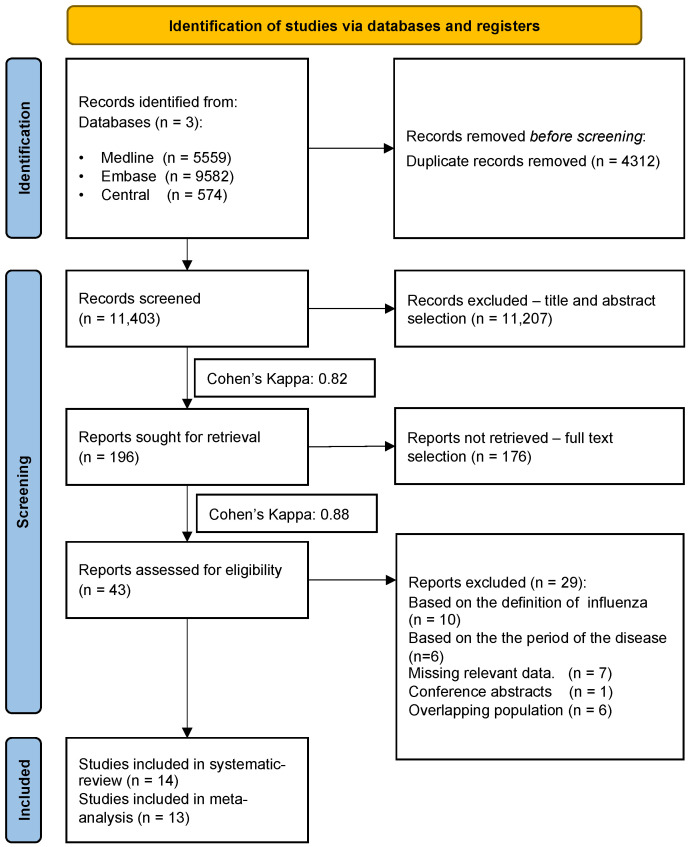
Flowchart of the systematic search and selection.

**Figure 2 viruses-14-02708-f002:**
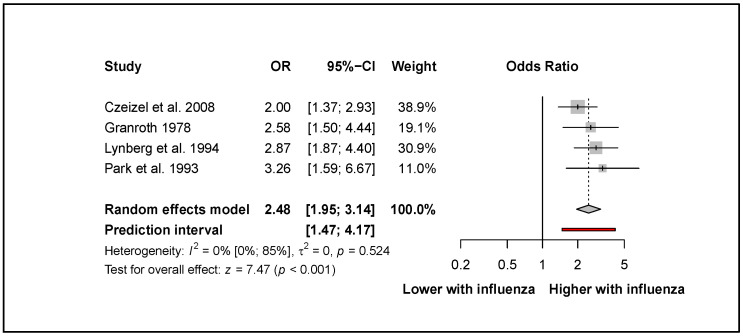
The odds of developing neural tube defects after influenza infection in the first trimester [34,35,36,38].

**Figure 3 viruses-14-02708-f003:**
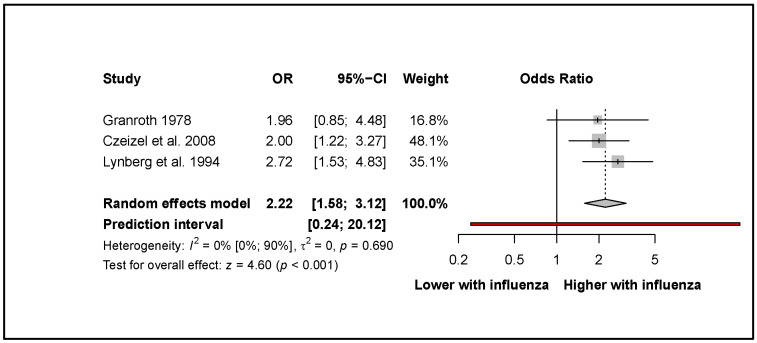
The odds of developing spina bifida after influenza infection in the first trimester [34,35,36].

**Figure 4 viruses-14-02708-f004:**
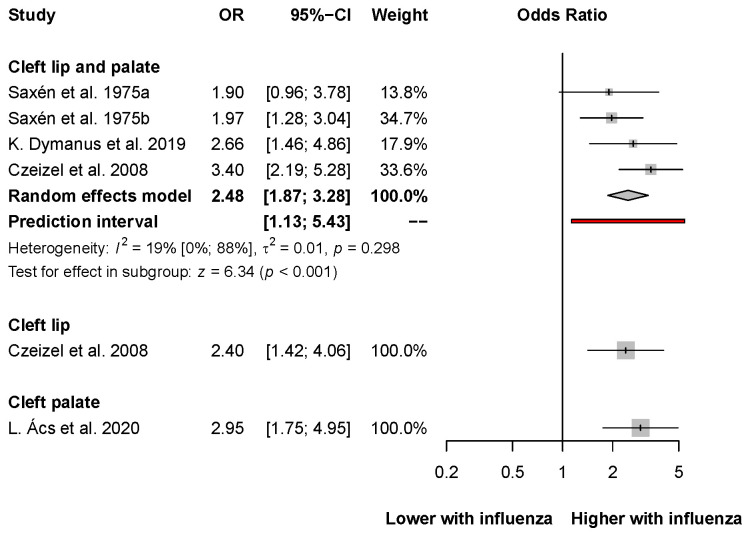
The odds of developing oral clefts after influenza infection in the first trimester [34,39,40,41].

**Figure 5 viruses-14-02708-f005:**
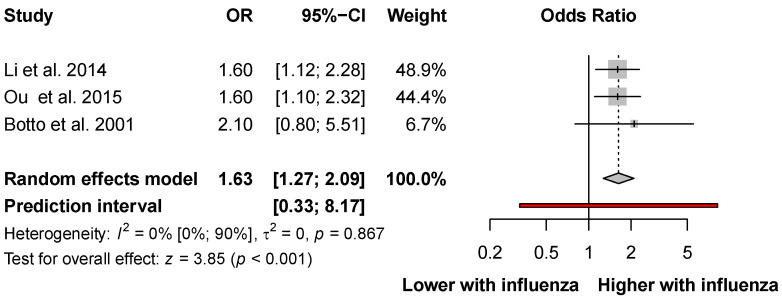
Adjusted odds ratio of developing congenital heart defects after influenza infection in the first trimester [8,32,37].

**Table 1 viruses-14-02708-t001:** Basic characteristics of included studies.

Author, Year	Study Type	Country	Number of the Participant Centers	Examination Period	Number of Patients (Cases-Controls)	Age Groups	Diagnosis of Influenza	Congenital Malformations	Adjusted for
L. Ács et al., 2020 [29]	Case-control	Hungary	No data	1980–1996, 2007–2009	1947 (751–1196)	<23; 23–33; >33	Prospective, medically recorded data; retrospective questionnaire, supplementary data collection	Cleft palate	
Ács et al., 2005 [30]	Case-control	Hungary	No data	1980–1996	3166 (1328–1838)	≤24; 25–29; >30	Prospective, medically recorded data; retrospective questionnaire, supplementary data collection	Neural tube defects; Cleft lip/palate; Cleft palate; Esophageal atresia; Pyloric stenosis; Intestinal atresia/stenosis; Rectal/anal stresia/stenosis; Renal a/dysgenesia; Obstructive urinary Cas; Hypospadiasis; Undescended testis; Exomphalos/gastroschisis; Congenital hydrocephaly, Ear CAs, Cardiovascular CAs, Clubfoot, Limb reduction defects, Poly/syndactilia, Diaphragmatic CAs, Other Isolated CAs, Multiple CAs	maternal employment status and use of antipyretic drugs
Aro et al., 1983 [31]	Case-control	Finland	No data	1964–1977	906 (453–453)	No data	Personal visits, questionnaire	limb reduction defects	
Botto et al., 2001 [32]	Case-control	USA	No data	1968–1980	3934 (905–3029)	11–19, 20–24, 25–29, 30–34, >35	Telephone interview	Congenital heart defects, Transposition of great arteries, Tetralogy of Fallot, Atrioventricular septal defect, Ebstein anomaly, Anomalous pulmonary venous return, All right obstructive defects, Tricuspid atresia, All left obstructive defects, Hypoplastic left heart, Aortic stenosis, Aortic coarctation, Ventricular septal defect, Atrial septal defect	maternal race, education, multivitamin use, smoking, alcohol use, chronic illnesses, and period of birth of the child
Busby et al., 2005 [33]		England	No data	1987–1994	275		General practitioner data, laboratory data	Anophtalmia, Micophtalmia	
Czeizel et al., 2008 [34]	Case-control	Hungary	No data	1980–1996	3754 (1349–2405)	No data	Prospective, medically recorded data; retrospective questionnaire, supplementray data collection	Neural-tube defects, Anencephalus+-spina bifida, Spina bifida aperta/cystica, Encephalocele, occipital, Microcephaly, primary, Congenital hydrocephalus, CAs of eye, Anophthalmia–microphthalmia, Primary congenital glaucoma, Congenital cataract, Ocular coloboma, CAs of ear, Auditory canal+ear Cas, An/microtia, Others, unspecified, Cardiovascular CAs, Transposition of great vessels, Ventricular septal defect, Atrial septal defect, type II, Hypoplastic left heart, Patent ductus arteriosus, CAs of aorta, CAs of pulmonary valve, Others or unspecified, Brachial cyst, cleft, fistula, preauricular sinus, CAs of respiratory system, Cleft palate, Robin sequence, Cleft lip+-cleft palate, Cleft lip, Cleft lip with palate, Esophageal atresia/stenosis with or without tracheoesophageal fistula, Cong hypertrophic pyloric stenosis, Atresia/stenosis of small intestine, Atresia/stenosis of rectum/anal canal, Other CAs of digestive system, Hirschprung’s disease, Other CAs of intestine, Other CAs of digestive system, Undescended testis (diagnosed after 3rd postnatal month), Hypospadias (without coronal type), Other CAs of genital organs, Renal a/dysgenesis, Obstructive CAs of urinary tract, Cystic kidney (diseases), Obstructive CAs of renal pelvis and ureter (hydronephrosis, constricture of ureteropelvic junction and ureterovesical orifice), Other CAs of urinary tract, Other CAs of kidney, Other CAs of bladder and urethrea, Clubfoot, Poly/syndactyly, Polydactyly, Syndactyly (without minor), Limb deficiencies, Other CAs of limbs, CAs of diaphragm, CAs of abdominal wall (exomphalos and gastroschisis are not differentiated), Multiple CAs (major gene mutations and chromosomal aberrations are excluded)	
Dymanus et al., 2019 [41]	Retrospective observational population study	USA	No data	2004–2013	58,270	No data		Cleft lip	
Granroth 1978 [35]	Case-control	Finland	No data	1965–1973	1420 (710–710)	No data	Questionnaire	Anencephalia, Spina bifida, Congenital hydrocephaly, Microcephaly, Hydrancephaly, Polydactylia	
Li et al., 2014 [8]	Case-control	China	No data	2010–2011	710 (294–416)	<20, 20–24, 25–29, 30–34, ≥35	Questionnaire	All congenital heart defects, Septal defects, Conotruncal defects, Right-sided obstructive defects, Left-sided obstructive defects, Anomalous pulmonary venous return, Other isolated CAs	maternal age, maternal education, maternal BMI, supplementation, and history of pregnancy with any defect
Lynberg et al., 1994 [36]	Case-control	USA	No data	1968–1980	329 (31–298)	No data	Questionnaire	Anencephalia, Spina bifida, Encephalocele	maternal age, education, smoking, alcohol consumption, and periconceptional multivitamin use
Ou et al., 2015 [37]	Case-control	China	39	2004–2013	8068 (4034–4034)	<30, 30–34, 35–40, >40	Questionnaire	Cardiovascular CAs, Ventricular septal defect, Atrial septal defect, Pulmonary stenosis, Dextro-transposition of great arteries, Tetralogy of Fallot	
Park et al., 1993 [38]	Case-control	USA	No data	1968–1980	1490 (304–1186)	11–19, 20–34, >35	Questionnaire	Anencephalia, Spina bifida	
Saxen et al., 1975 [39]	Case-control	Finland	No data	1972–1973	388 (194–194)	≥30 (23%)	Maternity records, interview records, death certificates	Cleft lip and palate	
Saxen et al., 1975 [40]	Case-control	Finland	No data	1967–1971	1198 (599–599)	≥30 (30.7%)	Maternity records, interview records, death certificates	Cleft lip and palate	

**Table 2 viruses-14-02708-t002:** Unadjusted odds ratios for developing specific types of congenital heart defects after influenza infection in the first trimester.

Study	Congenital Heart Defects	Odds Ratios
Li et al., 2014 [8]	Anomalous pulmonary venous return	0.80 (CI: 0.36−1.79)
Czeizel et al., 2008 [34]	Atrial septal defect	1.70 (CI: 0.42−6.96)
Czeizel et al., 2008 [34]	**CAs of the aorta**	**4.60 (CI: 1.41−15.01)**
Czeizel et al., 2008 [34]	CAs of the pulmonary valve	1.20 (CI: 0.38−3.75)
Li et al., 2014 [8]	**Conotruncal defects**	**1.71 (CI: 1.13−2.58)**
Czeizel et al., 2008 [34]	Hypoplastic left heart	1.60 (CI: 0.31−8.26)
Li et al., 2014 [8]	Left−sided obstructive heart defects	1.50 (CI: 0.84−2.68)
Czeizel et al., 2008 [34]	Patent ductus arteriosus	0.80 (CI: 0.08−7.55)
Li et al., 2014 [8]	Right−sided obstructive heart defects	1.40 (CI: 0.80−2.45)
Li et al., 2014 [8]	**Septal defects**	**1.92 (CI: 1.31−2.82)**
Czeizel et al., 2008 [34]	Transposition of great vessels	2.90 (CI: 0.90−9.32)
Czeizel et al., 2008 [34]	**Ventricular septal defect**	**2.70 (CI: 1.59−4.58)**

**Table 3 viruses-14-02708-t003:** Detailed odds ratios for developing specific congenital heart defects after influenza in the first trimester.

**Study**	**Congenital Heart Defects**	**Adjusted Odds Ratio**	**Adjusted for**	**Type of Logistic Regression**
Botto et al., 2001 [32]	Anomalous pulmonary venous return	2.20 (CI: 0.29−16.51)	maternal race, education, multivitamin use, smoking, alcohol use, chronic illnesses, and period of birth of the child	Multiple
Botto et al., 2001 [32]	**Aortic coarctation**	**3.80 (CI: 1.62−8.91)**		
Botto et al., 2001 [32]	Aortic stenosis	4.00 (CI: 0.90−17.84)		
Botto et al., 2001 [32]	Atrial septal defect	1.00 (CI: 0.12−8.60)		
Botto et al., 2001 [32]	Ebstein anomaly	3.00 (CI: 0.39−23.19)		
Botto et al., 2001 [32]	Hypoplastic left heart	1.60 (CI: 0.39−6.55)		
Botto et al., 2001 [32]	**Left−sided obstructive heart defects**	**2.90 (CI: 1.49−5.65)**		
Botto et al., 2001 [32]	**Right−sided obstructive heart defects**	**2.50 (CI: 1.14−5.49)**		
Botto et al., 2001 [32]	Septal defects	2.00 (CI: 0.38−14.28)		
Botto et al., 2001 [32]	Tetralogy of Fallot	0.50 (CI: 0.08−3.00)		
Botto et al., 2001 [32]	Transposition of great vessels	2.10 (CI: 0.80−5.51)		
Botto et al., 2001 [32]	Tricuspid atresia	7.90 (CI: 0.80−78.47)		
Botto et al., 2001 [32]	**Ventricular septal defect**	**2.00 (CI: 1.11−3.62)**		
Li et al., 2014 [8]	Anomalous pulmonary venous return	0.68 (CI: 0.28−1.65)	maternal age, maternal education, maternal body mass index (BMI), supplementation, and history of pregnancy with any defect	Multivariate
Li et al., 2014 [8]	**Conotruncal defects**	**1.60 (CI: 1.01−2.52)**		
Li et al., 2014 [8]	Left−sided obstructive heart defects	1.55 (CI: 0.82−2.93)		
Li et al., 2014 [8]	**Septal defects**	**2.12 (CI: 1.38−3.26)**		
Li et al., 2014 [8]	Right−sided obstructive heart defects	1.26 (CI: 0.68−2.34)		
Ou et al., 2015 [37]	**Atrial septal defect**	**2.71 (CI: 1.17−6.33)**	maternal age, maternal education, maternal pregnancy history, maternal environmental risk exposures, maternal perinatal diseases, and medication use in the first trimester	Multivariate
Ou et al., 2015 [37]	Pulmonary stenosis	0.62 (CI: 0.07−5.58)		
Ou et al., 2015 [37]	Transposition of great vessels	1.15 (CI: 0.07−19.22)		
Ou et al., 2015 [37]	Ventricular septal defect	1.19 (CI: 0.71−1.99)		

## Data Availability

The datasets used in this study can be found in the full-text articles included in the systematic review and meta-analysis.

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
