# Peer review of "First-Trimester Influenza Infection Increases the Odds of Non-Chromosomal Birth Defects: A Systematic Review and Meta-Analysis"

_viruses, 2022, doi:10.3390/v14122708_

Round 1
Reviewer 1 Report
The manuscript by Ákos Mátrai et al. reports the results of a systematic review and meta-analysis on the association between maternal influenza during the first trimester of pregnancy and all types of non-chromosomal birth defects. The authors performed the systematic search on 20th October, 2021.and overall 14 articles were included in the systemic review and 13 studies in meta-analysis. The performed steps and analyses are well-conducted and described in detail.
The study’s findings suggest that maternal influenza in the first trimester may increase the odds of non-chromosomal birth defects. The question raised by this study is important since only one meta-analysis exists on this topic. However, that paper includes studies which do not restrict the period of maternal influenza to the first trimester of pregnancy, i.e., to the critical period for the development of birth defects.
I have some questions that need to be addressed before the paper can be accepted for publication.
- Authors state that overlapping populations were excluded from the analyses. Table 1 describes the included 14 studies, and from that list it seems that two studies were based on the same case-control dataset (Ács et al., 2005 and Czeizel et al., 2008).
- As described in Material and Methods section, authors aimed to perform a meta-analysis if there were at least three studies for one outcome. However, the forest plot on specific types of neural tube defects (Figure S1) presents the results of a random effect model also in case of two included studies (e.g., Encephalocele).
- The meta-analysis includes 13 case-control studies which are prone to recall bias. I would recommend highlighting this and describing the effect of this type of bias in more details in the Discussion section (especially discussing how this bias may affect the conclusion).
- The section which describes the meta-analysis says that p-value less than 0.05 was considered statistically significant. Was the conclusion of the study based on statistical significance testing or the authors used confidence intervals to evaluate their findings? (please see articles: Wasserstein (2019). Moving to a world beyond “p< 0.05”. The American Statistician, 73(sup1), 1-19. and Nature (https://www.nature.com/articles/d41586-019-00857-9; Amrhein, V. (2019). Scientists rise up against statistical significance.))
- I suggest checking the abbreviations through the manuscript – sometimes they are used before their definition (e.g. CA in introduction).
- Table 1 interchanges the content of columns “Country” and “Examination period” in some cases (e.g. Park et al, 1993).
- Suggest improving the language of the manuscript (e.g., 1. title of Figure 5: “Adjusted odds of developing congenital heart defects…” -> “Adjusted odds ratio of developing congenital heart defects…; 2. Abstract: “highlight the risk of influenza infection during the first trimester of pregnancy.” -> “highlight the risk of birth defect after influenza infection during the first trimester of pregnancy.”; 3. Results/Congenital heart defects: “when the analysis was performed with multiple logistic regression” – It is not clear whether the authors mean multiple or multivariable regression model?
- Authors present a positive association between influenza and specific types of congenital heart defects (Results section / Congenital heart defects). These results are presented in Table 4, however it would be beneficial to see the exact point estimates and belonging 95% CIs also in the text.
Reviewer 2 Report
In this manuscript, Matrai et al. have conducted a systematic review and meta-analysis of from a large number of published reports on the birth defects due to maternal influenza in the first trimester of pregnancy. Standard statistical tests were used to confirm significance of the effects, resulting in the conclusion that first-trimester maternal influenza is associated with increased odds of developing all types of birth defects, including some serious ones, such as neural tube defects or cleft lip and palate, and heart abnormalities. Overall, the work is comprehensive and well-done, generating clinically important findings. I have only a few minor comments, as follows.
As the authors mentioned, as least one study (Ref. 6; Lutejin et al. 2014) collected data on the full pregnancy period. It would be interesting to know those results were similar to the authors’. If so, it will strengthen the presumption of Matrai et al. that the early-stage fetus is indeed more prone to flu-induced defects. The authors may just discuss this briefly without presenting any data, or they can state why the comparison may not be meaningful.
Line 27: Please change “found” to “considered”.
Reviewer 3 Report
Welldone on your submission. Please see below my comments:
Keywords
1) Please use another set of keywords that are not included in the study title.
Introduction
2) The introduction section is poorly written and should be rewritten.
3) The third paragraph of the introduction is quite disjointed. Starting the paragraph with "On the other hand, the prevention of CAs is considered....." does not make sense because there was no previous discussion on CAs. Please define CA at first mention in line 75.
4) Try linking the four sentences below so they make logical sense:
"Influenza may occur in pregnant women [3]. Organogenesis takes place in the first trimester, so any environmental effects that occur during this period affect the embryo. However, the role of influenza viruses is debated in the origin of congenital abnormalities (CAs). Worldwide, 3-5% of newborns are affected by a congenital anomaly [5]".
5) Please provide some background information on CA and how they relate to influenza virus.
6) The buildup to study rationale is weak. Provide a more robust rationale for the conduct of the study.
7) Provide references for the epidemiologic studies mentioned in introduction.
8) The study aim in the abstract differs from that in the introduction section. Please ensure the study aims reads the same both in the abstract and the introduction.
Materials and Methods
9) The last search was performed 20th October 2021. This is more than one year ago, was a rerun of the search done recently incase new studies on the topic have been conducted? Given the search was last run more than a year ago, I strongly recommend a rerun of the search.
10) What types of studies were included in the review? were grey literature included?
11) The inclusion/exclusion criteria are not adequately communicated.
12)"When the study population was overlapping for an outcome, we included the study with a larger sample size". What do you mean by this? Since this is a systematic review, all studies should be included irrespective of sample size.
Round 2
Reviewer 1 Report
The authors have answered adequately all my questions. I do not have further comments.